# ON MEANING-PRESERVING ADVERSARIAL PERTURBATIONS FOR SEQUENCE-TO-SEQUENCE MODELS

## ABSTRACT

Adversarial examples have been shown to be an effective way of assessing the robustness of neural sequence-to-sequence (seq2seq) models, by applying perturbations to the input of a model leading to large degradation in performance. However, these perturbations are only indicative of a weakness in the model if they do not change the semantics of the input in a way that would change the expected output. Using the example of machine translation (MT), we propose a new evaluation framework for adversarial attacks on seq2seq models taking meaning preservation into account and demonstrate that existing methods may not preserve meaning in general. Based on these findings, we propose new constraints for attacks on word-based MT systems and show, via human and automatic evaluation, that they produce more semantically similar adversarial inputs. Furthermore, we show that performing adversarial training with meaning-preserving attacks is beneficial to the model in terms of adversarial robustness without hurting test performance.

## 1 INTRODUCTION

Attacking a machine learning model with adversarial perturbations is the process of making changes to its input to maximize an adversarial goal, such as mis-classification (Szegedy et al., 2013) or mis-translation (Zhao et al., 2018). These attacks provide insight into the vulnerabilities of machine learning models and their brittleness to samples outside the training distribution. This is critical for systems sensitive to safety or security, *e.g.* self-driving cars (Bojarski et al., 2016).

Adversarial attacks were first defined and investigated for computer vision systems (Szegedy et al. (2013); Goodfellow et al. (2014); Moosavi-Dezfooli et al. (2016) inter alia), where they benefit from the fact that images are expressed in continuous space, making minuscule perturbations largely imperceptible to the human eye. In discrete spaces such as natural language sentences, the situation is more problematic; even a flip of a single word or character is generally perceptible by a human reader. Thus, most of the mathematical framework in previous work is not directly applicable to discrete text data. Moreover, there is no canonical distance metric for textual data like the $\ell_2$ norm in real-valued vector spaces such as images, and evaluating the level of semantic similarity between two sentences is a field of research of its own (Cer et al., 2017). This elicits a natural question: what does the term "adversarial perturbation" mean in the context of natural language processing (NLP)?

We propose a simple but natural criterion for adversarial examples in NLP, particularly seq2seq models: adversarial examples should be meaning-preserving on the source side, but meaning-destroying on the target side. The focus on explicitly evaluating meaning preservation is in contrast to previous work on adversarial examples for seq2seq models (Belinkov & Bisk, 2018; Zhao et al., 2018; Cheng et al., 2018; Ebrahimi et al., 2018a). Nonetheless, this feature is extremely important; given two sentences with equivalent meaning, we would expect a good model to produce two outputs with equivalent meaning.In other words, any meaning-preserving perturbation that results in the model output changing drastically highlights a fault of the model.

A first technical contribution of the paper is to lay out a method for formalizing this concept of meaning-preserving perturbations (§2). This makes it possible to evaluate the effectiveness of adversarial attacks or defenses either using gold-standard human evaluation, or approximations that can be calculated without human intervention. We further propose a simple method of imbuing gradient-

based word substitution attacks (§3.1) with simple constraints aimed at increasing the chance that the meaning is preserved (§3.2).

Our experiments are designed to answer several questions about meaning preservation in seq2seq models. First, we evaluate our proposed "source-meaning-preserving, target-meaning-destroying" criterion for adversarial examples using both manual and automatic evaluation (§4.2) and find that a less widely used evaluation metric (chrF) provides significantly better correlation with human judgments than the more widely used BLEU and METEOR metrics. We proceed to perform an evaluation of adversarial example generation techniques, finding that constrained substitution attacks do preserve meaning to a higher degree than unconstrained attacks while still degrading the performance of the systems across different languages and model architectures (§4.3). Finally we apply existing methods for adversarial training to the adversarial examples with these constraints and show that making adversarial inputs more semantically similar to the source is beneficial for robustness to adversarial attacks and does not decrease test performance (§5).

## 2 A FRAMEWORK FOR EVALUATING ADVERSARIAL ATTACKS

In this section, we present a simple procedure for evaluating adversarial attacks on seq2seq models. We will use the following notation: $x$ and $y$ refer to the source and target sentence respectively. We denote $x$'s translation by model $M$ as $y_M(x)$. Finally, $\hat{x}$ and $y_M(\hat{x})$ represent an adversarially perturbed version of $x$ and its translation by $M$, respectively. The nature of $M$ and the procedure for obtaining $\hat{x}$ from $x$ are irrelevant to the discussion below.

### 2.1 THE ADVERSARIAL TRADE-OFF

The goal of adversarial perturbations is to produce failure cases for the model $M$. Hence, the evaluation must include some measure of the target similarity between $y$ and $y_M(\hat{x})$, which we will denote $s_{tgt}(y, y_M(\hat{x}))$. However, if no distinction is being made between perturbations that preserve the meaning and those that don't, a sentence like "he's very *friendly*" is considered a valid adversarial perturbation of "he's very *adversarial*", even though its meaning is the opposite. Hence, it is crucial, when evaluating adversarial attacks on MT models, that the discrepancy between the original and adversarial input sentence be quantified in a way that is sensitive to meaning. Let us denote such a source similarity score $s_{src}(x, \hat{x})$.

Based on these functions, we define the target relative score decrease as:

$$d_{tgt}(y, y_M(x), y_M(\hat{x})) := \begin{cases} 0 & \text{if } s_{tgt}(y, y_M(\hat{x})) \geq s_{tgt}(y, y_M(x)) \\ \frac{s_{tgt}(y, y_M(x)) - s_{tgt}(y, y_M(\hat{x}))}{s_{tgt}(y, y_M(x))} & \text{otherwise} \end{cases} \quad (1)$$

The choice to report the relative decrease in $s_{tgt}$ makes scores comparable across different models or languages[1]. For instance, for languages that are comparatively easy to translate (*e.g.* French-English), $s_{tgt}$ will be higher in general, and so will the gap between $s_{tgt}(y, y_M(x))$ and $s_{tgt}(y, y_M(\hat{x}))$. However this does not necessarily mean that attacks on this language pair are more effective than attacks on a "harder" language pair (*e.g.* Czech-English) where $s_{tgt}$ is usually smaller.

We recommend that both $s_{src}$ and $d_{tgt}$ be reported when presenting adversarial attack results. However, in some cases where a single number is needed, we suggest reporting the attack's success $\mathcal{S} := s_{src} + d_{tgt}$. The interpretation is simple: $\mathcal{S} > 1 \Leftrightarrow d_{tgt} > 1 - s_{src}$, which means that the attack has destroyed the target meaning ($d_{tgt}$) more than it has destroyed the source meaning ($1 - s_{src}$).

### 2.2 SIMILARITY METRICS

Throughout §2.1, we have not given an exact description of the semantic similarity scores $s_{src}$ and $s_{tgt}$. Indeed, automatically evaluating the semantic similarity between two sentences is an open area of research and it makes sense to decouple the definition of adversarial examples from the specific method used to measure this similarity. In this section, we will discuss manual and automatic metrics that may be used to calculate it.

---

[1] Note that we do not allow negative $d_{tgt}$ to keep all scores between 0 and 1.

### 2.2.1 HUMAN JUDGMENT

Judgment by speakers of the language of interest is the *de facto* gold standard metric for semantic similarity and specific criteria for such as adequacy/fluency (Ma & Cieri, 2006), acceptability (Goto et al., 2013), and 6-level semantic similarity (Cer et al., 2017) have been used in evaluations of MT and sentence embedding methods. In the context of adversarial attacks, we propose the following 6-level evaluation scheme, which is motivated by previous measures, but designed to be (1) symmetric, like Cer et al. (2017), (2) and largely considers meaning preservation but at the very low and high levels considers fluency of the output[2], like Goto et al. (2013):

---

How would you rate the similarity between the meaning of these two sentences?

    0. The meaning is completely different or one of the sentences is meaningless

    1. The topic is the same but the meaning is different

    2. Some key information is different

    3. The key information is the same but the details differ

    4. Meaning is essentially equal but some expressions are unnatural

    5. Meaning is essentially equal and the two sentences are well-formed English

---

### 2.2.2 AUTOMATIC METRICS

Unfortunately, human evaluation is expensive, slow and sometimes difficult to obtain, for example in the case of low-resource languages. This makes automatic metrics that do not require human intervention appealing for experimental research. This section describes 3 evaluation metrics commonly used as alternatives to human evaluation, in particular to evaluate translation models.[3]

**BLEU:** (Papineni et al., 2002) is an automatic metric based on n-gram precision coupled with a penalty for shorter sentences. It relies on exact word-level matches and therefore cannot detect synonyms or morphological variations.

**METEOR:** (Denkowski & Lavie, 2014) first estimates alignment between the two sentences and then computes unigram F-score (biased towards recall) weighted by a penalty for longer sentences. Importantly, METEOR uses stemming, synonymy and paraphrasing information to perform alignments. On the downside, it requires language specific resources.

**chrF:** (Popović, 2015) is based on the character $n$-gram F-score. In particular we will use the chrF2 score (based on the F2-score – recall is given more importance), following the recommendations from Popović (2016). By operating on a sub-word level, it can reflect the semantic similarity between different morphological inflections of one word (for instance), without requiring language-specific knowledge which makes it a good one-size-fits-all alternative.

Because multiple possible alternatives exist, it is important to know which is the best stand-in for human evaluation. To elucidate this, we will compare these metrics to human judgment in terms of Pearson correlation coefficient on outputs resulting from a variety of attacks in §4.2.

## 3 GRADIENT-BASED ADVERSARIAL ATTACKS

In this section, we overview the adversarial attacks we will be considering in the rest of this paper.

### 3.1 ATTACK PARADIGM

We perform gradient-based attacks that replace one word in the sentence so as to maximize an adversarial loss function $\mathcal{L}_{\mathrm{adv}}$, similar to the substitution attacks proposed in (Ebrahimi et al., 2018b).

---

[2]This is important to rule out nonsensical sentences and distinguish between clean and "noisy" (*e.g.* typos, non-native speech. . . ) paraphrases.

[3]Note that other metrics of similarity are certainly applicable within the overall framework of §2.2.1, but we limit our examination in this paper to the three noted here.

### 3.1.1 GENERAL APPROACH

Precisely, for a word-based translation model $M$,[4] and given an input sentence $w_1, \ldots, w_n$, we find the position $i^*$ and word $w^*$ satisfying the following optimization problem:

$$\arg\max_{1 \leq i \leq n, \hat{w} \in \mathcal{V}} \mathcal{L}_{\text{adv}}(w_0, \ldots, w_{i-1}, \hat{w}, w_{i+1}, \ldots, w_n) \tag{2}$$

Where $\mathcal{L}_{\text{adv}}$ is a differentiable function which represents our adversarial objective. Using the first order approximation of $\mathcal{L}_{\text{adv}}$ around the original word vectors $\mathbf{w}_1, \ldots, \mathbf{w}_n$[5], this can be derived to be equivalent to optimizing $\arg\max_{1 \leq i \leq n, \hat{w} \in \mathcal{V}} [\hat{\mathbf{w}} - \mathbf{w}_i]^\mathsf{T} \nabla_{\mathbf{w}_i} \mathcal{L}_{\text{adv}}$.

The above optimization problem can be solved by brute-force in $\mathcal{O}(n|\mathcal{V}|)$ space complexity, whereas the time complexity is bottlenecked by a $|\mathcal{V}|d$ times $nd$ matrix multiplication, which is not more computationally expensive than computing logits during the forward pass of the model. Overall, this naive approach is sufficiently fast to be conducive to adversarial training. We also found that the attacks benefited from normalizing the gradient by taking its sign.

Extending this approach to finding the optimal perturbations for more than 1 substitution would require exhaustively searching over all possible combinations. However previous work (Ebrahimi et al., 2018a) suggests that greedy search is a good enough approximation.

### 3.1.2 THE ADVERSARIAL LOSS $\mathcal{L}_{\text{ADV}}$

We want to find an adversarial input $\hat{x}$ such that, assuming that the model has produced the correct output $y_1, \ldots, y_{t-1}$ up to step $t-1$ during decoding, the probability that the model makes an error at the next step $t$ is maximized. In the log-semiring, this translates into the following loss function:

$$\mathcal{L}_{\text{adv}}(\hat{x}, y) = \sum_{t=1}^{m} \log(1 - p(y_t \mid \hat{x}, y_1, \ldots, y_{t-1})) \tag{3}$$

### 3.2 ENFORCING SEMANTICALLY SIMILAR ADVERSARIAL INPUTS

As a contrast to these previous methods, which don't consider meaning preservation, we propose simple modifications of the approach presented in §3.1 to create adversarial perturbations that are more likely to preserve meaning. The basic idea is to restrict the possible word substitutions to similar words. We compare two sets of constraints:

**kNN:** This constraint enforces that the word be replaced only with one of its 10 nearest neighbors in the source embedding space. This has two effects: first, the replacement will be likely semantically related to the original word (if words close in the embedding space are indeed semantically related, as hinted by Table 1). Second, it ensures that the replacement's word vector is close enough to the original word vector that the first order assumption is more likely to be satisfied.

**CharSwap:** This constraint requires that the substituted words must be obtained by swapping characters. Word internal character swaps have been shown to not affect human readers greatly (McCusker et al., 1981), hence making them likely to be meaning-preserving. Moreover we add the additional constraint that the substitution must not be in the vocabulary, which will likely be particularly meaning-destroying on the target side for the word-based models we test here. In such cases where word-internal character swaps are not possible or can't produce out-of-vocabulary (OOV) words, we resort to the naive strategy of repeating the last character of the word. The exact procedure used to produce this kind of perturbations is described in appendix A.1.

In contrast, we refer the base attack without constraints as **Unconstrained** hereforth. Table 1 gives qualitative examples of the kind of perturbations generated under the different constraints.

---

[4]Alternatively, we could have chosen character-based (Lee et al., 2017) or subword-based (Sennrich et al., 2016) models. For character-based models the formulation would have been largely similar. For subword-based models, additional difficulty would be introduced due to changes to the input resulting in different subword segmentations. While these challenges are interesting, they are beyond the scope of the current work.

[5]More generally we will use the bold $\mathbf{w}$ when talking about the embedding vector of word $w$

Table 1: Examples of different adversarial inputs. The substituted word is highlighted.

| | |
|---|---|
| Original | **Pourquoi** faire cela ? |
| English gloss | **Why** do this? |
| Unconstrained | **construisant** (English: building) faire cela ? |
| kNN | **interrogez** (English: interrogate) faire cela ? |
| CharSwap | **Puorquoi** (typo) cela ? |
| Original | Si seulement je pouvais me muscler **aussi** rapidement. |
| English gloss | If only I could build my muscle **this** fast. |
| Unconstrained | Si seulement je pouvais me muscler **etc** rapidement. |
| kNN | Si seulement je pouvais me muscler **plsu** (typo for "more") rapidement. |
| CharSwap | Si seulement je pouvais me muscler **asusi** (typo) rapidement. |

## 4 EXPERIMENTS

Our experiments serve two purposes. First, we examine our proposed framework of evaluating adversarial attacks (§2), and also elucidate which automatic metrics correlate better with human judgment for the purpose of evaluating adversarial attacks (§4.2). Second, we use this evaluation framework to compare various adversarial attacks and demonstrate that adversarial attacks that are explicitly constrained to preserve meaning receive better assessment scores (§4.3).

### 4.1 EXPERIMENTAL SETTING

**Data:** Following previous work on adversarial examples for seq2seq models (Belinkov & Bisk, 2018; Ebrahimi et al., 2018a), we perform all experiments on the IWSLT2016 dataset (Cettolo et al., 2016) in the {French,German,Czech}→English directions (`fr-en`, `de-en` and `cs-en`). We compile all previous IWSLT test sets before 2015 as validation data, and keep the 2015 and 2016 test sets as test data. The data is tokenized with the Moses tokenizer (Koehn et al., 2007). The exact data statistics can be found in Appendix A.2. We only keep the 40,000 most frequent word and replace all others with `<unk>` tokens.

**MT Models:** We perform experiments with two common neural machine translation (NMT) models. The first is an LSTM based encoder-decoder architecture with attention (Luong et al., 2015). It uses 2-layer encoders and decoders, and dot-product attention. We set the word embedding dimension to 300 and all others to 500. The second model is a self-attentional Transformer (Vaswani et al., 2017), with 6 1024-dimensional encoder and decoder layers and 512 dimensional word embeddings. Both the models are trained with Adam (Kingma & Ba, 2014), dropout (Srivastava et al., 2014) of probability 0.3 and label smoothing (Szegedy et al., 2016) with value 0.1. During inference, we replace `<unk>` tokens in the translated sentences with the source words with the highest attention value. The full experimental setup and source code are at `redacted_url`.

**Automatic Metric Implementations:** To evaluate both sentence and corpus level BLEU score, we first de-tokenize the output and use `sacreBLEU`[6] (Post, 2018) with its internal `intl` tokenization, to keep BLEU scores agnostic to tokenization. We compute METEOR using the official implementation[7]. ChrF is reported with the `sacreBLEU` implementation on detokenized text with default parameters.

### 4.2 CORRELATION OF AUTOMATIC METRICS WITH HUMAN JUDGMENT

We first examine which of the automatic metrics listed in §2.2 correlates more with human judgment for our adversarial attacks. For this experiment, we restrict the scope to the case of the BiLSTM model on `fr-en`. For the French side, we randomly select 900 sentence pairs $(x, \hat{x})$ from the validation set, 300 for each of the constraints Unconstrained, kNN and CharSwap. To vary the level of perturbation, the 300 pairs contain an equal amount of perturbed input obtained by substituting 1, 2 and 3 words. On the English side, we select 900 pairs of reference translations and translations of adversarial input $(y, y_M(\hat{x}))$ with the same distribution of attacks as the source side, as well as

---

[6]`https://github.com/mjpost/sacreBLEU`
[7]`http://www.cs.cmu.edu/~alavie/METEOR/`

300 $(y, y_M(x))$ pairs (to include translations from original inputs). This amounts to 1,200 sentence pairs in the target side.

These sentences are sent to English and French speaking annotators to be rated according to the guidelines described in §2.2.1. Each sample (a pair of sentences) is rated by two independent evaluators. If the two ratings differ, the sample is sent to a third rater (an auditor and subject matter expert) who makes the final decision.

Finally, we compare the human results to each automatic metric with Pearson's correlation coefficient. The correlations are reported in Table 2. As evidenced by the results, chrF exhibits higher correlation with human judgment, followed by METEOR and BLEU. This is true both on the source side ($x$ vs $\hat{x}$) and in the target side ($y(x)$ vs $y_M(\hat{x})$). We evaluate the statistical significance of this result using a paired bootstrap test for $p < 0.01$. Notably we find that chrF is significantly better than METEOR in French but not in English. This is not too unexpected because METEOR has access to more language-dependent resources in English (specifically synonym information) and thereby can make more informed matches of these synonymous words and phrases. Moreover the French source side contains more "character-level" errors (from CharSwap attacks) which are not picked-up well by word-based metrics like BLEU and METEOR.

Table 2: Correlation of automatic metrics to human judgment of adversarial source and target sentences. "*" indicates that the correlation is significantly better than the next-best one.

| Language | BLEU | METEOR | chrF |
|---|---|---|---|
| French | 0.415 | 0.440 | 0.586* |
| English | 0.357 | 0.478* | 0.497 |

Thus, in the following sections, we report attack results both in terms of chrF in the source ($s_{src}$) and relative decrease in chrF (RDchrF) in the target ($d_{tgt}$).

## 4.3 ATTACK RESULTS

We can now compare attacks under the three constraints Unconstrained, kNN and CharSwap and draw conclusions on their capacity to preserve meaning in the source and destroy it in the target. Attacks are conducted on the validation set using the approach described in §3.1 with 3 substitutions (this means that each adversarial input is at edit distance at most 3 from the original input). Results are reported in Table 3 (on a scale of 1 to 100 for readability). To give a better idea of how the different variables (language pair, model, attack), we give a graphical representation of these same results in Figure 1. The rest of this section discusses the implication of these results.

Table 3: Target RDchrF and source chrF scores for all the attacks.

| Model | | BiLSTM | | | Transformer | | |
|---|---|---|---|---|---|---|---|
| Language pair | | cs-en | de-en | fr-en | cs-en | de-en | fr-en |
| | Original chrF | 45.68 | 49.43 | 57.49 | 47.66 | 51.08 | 58.04 |
| Target | Unconstrained | 25.37% | 25.53% | 25.59% | 25.24% | 24.99% | 24.68% |
| | CharSwap | 24.11% | 24.93% | 23.60% | 21.58% | 23.23% | 21.74% |
| | kNN | 14.99% | 15.59% | 15.22% | 20.73% | 19.96% | 18.58% |
| Source | Unconstrained | 70.13 | 72.39 | 74.29 | 69.03 | 71.93 | 73.22 |
| | CharSwap | 82.65 | 84.40 | 86.62 | 84.12 | 85.97 | 87.01 |
| | kNN | 78.08 | 78.10 | 77.62 | 74.94 | 77.92 | 77.88 |

**Adding Constraints Helps Preserve Source Meaning:** Comparing the kNN and CharSwap rows to Unconstrained in the "source" section of Table 3 clearly shows that constrained attacks have a positive effect on meaning preservation. Beyond validating our assumptions from §3.2, this shows that source chrF is useful to carry out the comparison in the first place. To give a point of reference, results from the manual evaluation carried out in §4.2 show that that 90% of the French sentence pairs to which humans gave a score of 4 or 5 in semantic similarity have a chrF $> 78$.

**Different Models are not Equal in the Face of Adversity:** Inspection of the target-side results yields several interesting observations. First, the high RDchrF of CharSwap is yet another indication of word-based model's known shortcomings when presented with words out of their training vocabulary, even with <unk>-replacement. Second, and perhaps more interestingly, Transformer models appear to be less robust to small embeddings perturbations (kNN attacks) compared to BiL-

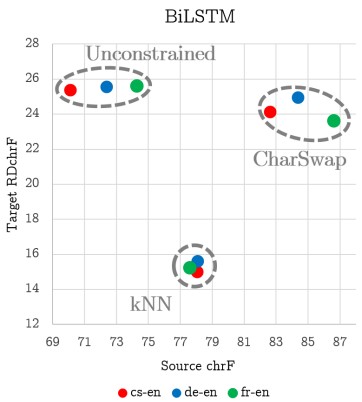 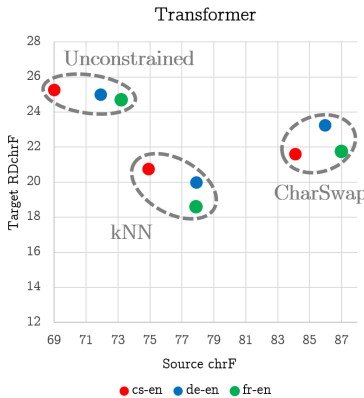

Figure 1: Graphical representation of the results in Table 3. High source chrF and target RDchrF (upper-right corner) indicates a good attack.

STMs. Although the exploration of the exact reasons for this phenomenon is beyond the scope of this work, this is a good example that RDchrF can shed light on the different behavior of different architectures when confronted with adversarial input. Overall, we find that the CharSwap constraint is the only one that consistently produces attacks with $> 1$ average success according to Table 3. Table 4 contains two qualitative examples of this attack on the BiLSTM in `fr-en`.

Table 4: Example of CharSwap attacks on the `fr-en` BiLSTM. The first example is a successful attack (high source chrF and target RDchrF) whereas the second is not.

| Successful attack (source chrF = 80.89, target RDchrF = 84.06) | |
| --- | --- |
| Original | Ils le réinvestissent directement en engageant plus de procès. |
| Adversarial source | Ilss le réinvestissent dierctement en engagaent plus de procès. |
| Reference | They plow it right back into filing more troll lawsuits. |
| Base output | They direct it directly by engaging more cases. |
| Adversarial output | .. de plus. |
| Unsuccessful attack (source chrF = 54.46, target RDchrF = 0.00) | |
| Original | C'était en Juillet 1969. |
| Adversarial source | C'étiat en Jiullet 1969. |
| Reference | This is from July, 1969. |
| Base output | This was in July 1969. |
| Adversarial output | This is. in 1969. |

# 5 ADVERSARIAL TRAINING WITH MEANING-PRESERVING ATTACKS

## 5.1 ADVERSARIAL TRAINING

Adversarial training augments the training data with adversarial examples. Formally, in place of the negative log likelihood (NLL) objective on a sample $x, y$, $\mathcal{L}_{NLL}(x, y) = NLL(x, y)$, the loss function is replaced with an interpolation of the NLL of the original sample $x, y$ and an adversarial sample $\hat{x}, y$: $\mathcal{L}'(x, y) = (1 - \alpha)NLL(x, y) + \alpha NLL(\hat{x}, y)$. Ebrahimi et al. (2018a) suggest that while adversarial training improves robustness to adversarial attacks, it can be detrimental to test performance on non-adversarial input. We investigate whether this is still the case when adversarial attacks are largely meaning-preserving.

In our experiments, we generate $\hat{x}$ (under the CharSwap constraint) on the fly at each training step. To maintain training speed we apply only one substitution to produce $\hat{x}$ from $x$. Although this results in "weaker" adversaries, we find that this makes training time less than $2\times$ slower than normal training. In addition, we compare adversarial training with adversarial perturbations chosen with the gradient-based approach from §3.1, CharSwap-adv to a baseline where the perturbation is chosen uniformly at random, CharSwap-rand.

Table 5: chrF (BLEU) scores on the original test set before/after adversarial training.

| Model | BiLSTM | | | Transformer | | |
|---|---|---|---|---|---|---|
| Language pair | cs-en | de-en | fr-en | cs-en | de-en | fr-en |
| Base | 44.21 | 49.30 | 55.67 | 45.95 | 50.77 | 56.41 |
| | (22.89) | (28.61) | (35.28) | (24.52) | (29.90) | (36.00) |
| CharSwap-adv | 44.15 | 49.20 | 55.72 | 45.92 | 50.73 | 56.21 |
| | (23.38) | (28.55) | (35.44) | (24.67) | (29.97) | (35.91) |
| CharSwap-rand | 44.07 | 49.14 | 55.79 | 46.04 | 50.75 | 56.72 |
| | (23.01) | (28.49) | (35.26) | (24.65) | (29.81) | (36.19) |

Table 6: Robustness to CharSwap attacks with/without adversarial training. Lower is better.

| Model | BiLSTM | | | Transformer | | |
|---|---|---|---|---|---|---|
| Language pair | cs-en | de-en | fr-en | cs-en | de-en | fr-en |
| Base | 24.11% | 24.93% | 23.60% | 21.58% | 23.23% | 21.74% |
| CharSwap-adv | 21.53% | 21.70% | 18.77% | 18.75% | 19.72% | 18.68% |
| CharSwap-rand | 22.29% | 22.48% | 19.80% | 19.35% | 19.32% | 17.49% |

## 5.2 RESULTS

Test performance on non-adversarial input is reported in Table 5. In keeping with the rest of the paper, we primarily report chrF results, but also show the standard BLEU as well. We observe no significant change in test performance on non-adversarial data. However, a look at the RDchrF of CharSwap attacks on the validation data (Table 6) shows that adversarial training clearly has a positive influence on robustness to adversarial attacks.

Hence, we can safely conclude that adversarial training with CharSwap attacks improves robustness while not impacting test performance in this setting. This is likely because CharSwap adversarial inputs are more constrained and less at risk of changing the meaning of the source sentence, thereby creating a spurious training sample $(\hat{x}, y)$ where $y$ is not an acceptable translation of $\hat{x}$.

## 6 RELATED WORK

Following seminal work on adversarial attacks by Szegedy et al. (2013), Goodfellow et al. (2014) introduced gradient-based attacks and adversarial training. Since then, a variety of attack (Moosavi-Dezfooli et al., 2016) and defense (Cissé et al., 2017; Kolter & Wong, 2017) mechanisms have been proposed. Adversarial examples for NLP specifically have seen attacks on sentiment (Papernot et al., 2016; Samanta & Mehta, 2017; Ebrahimi et al., 2018b), malware (Grosse et al., 2016), gender (Reddy & Knight, 2016) or toxicity (Hosseini et al., 2017) classification to cite a few.

In MT methods have been proposed to attack word-based (Zhao et al., 2018; Cheng et al., 2018) and character-based (Belinkov & Bisk, 2018; Ebrahimi et al., 2018a) models. However these works side-step the question of meaning preservation in the source: they mostly focus on target side evaluation. Finally there is work centered around meaning-preserving adversarial attacks for NLP via paraphrase generation (Iyyer et al., 2018) or rule-based approaches (Jia & Liang, 2017; Ribeiro et al., 2018; Naik et al., 2018). However the proposed attacks are highly engineered and focused on English.

## 7 CONCLUSION

This paper highlights the performance of meaning-preserving adversarial perturbations for NLP models (with a focus on seq2seq). We proposed a general evaluation framework for adversarial perturbations and compared various automatic metrics as alternatives to human judgment to instantiate this framework. We then confirmed that, in the context of MT, "naive" attacks do not preserve meaning in general, and proposed alternatives to remedy this issue. Finally, we have shown the utility of adversarial training in this paradigm. We hope that this helps future work in this area of research to evaluate meaning conservation more consistently.

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

## A    SUPPLEMENTAL MATERIAL

### A.1    GENERATING OOV REPLACEMENTS WITH INTERNAL CHARACTER SWAPS

```python
def make_oov(word, vocab, max_scrambling):
    """Modify a word to make it OOV (while keeping the meaning)"""
    # If the word has more than 3 letters try scrambling them
    if len(word) > 3:
        # For a fixed number of steps
        for _ in range(max_scrambling):
            # Swap two adjacent characters in the middle of the word
            pos = random.randint(1, len(word) - 3)
            word = word[:pos] + word[pos+1] + word[pos] + word[pos+2:]
            # If we got an OOV already just return it
            if word not in vocab:
                return word
    # If nothing worked, or the word is too short for scrambling,
    # just repeat the last character ad nauseam
    char = word[-1]
    while word in vocab:
        word = word + char
    return word
```

### A.2    IWSLT2016 DATASET

See table 7 for statistics on the size of the IWSLT2016 corpus used in our experiments.

|         | #train | #valid | #test |
|---------|--------|--------|-------|
| fr-en   | 220.4k | 6,824  | 2,213 |
| de-en   | 196.9k | 11,825 | 2,213 |
| cs-en   | 114.4k | 5,716  | 2,213 |

Table 7: IWSLT2016 data statistics.

