# OpenReview forum: "On Meaning-Preserving Adversarial Perturbations for Sequence-to-Sequence Models"
_ICLR.cc/2019/Conference_

### Official Review · AnonReviewer2 · 2018-10-31
**An inspiring study on adversarial attacks for natural language**

**Rating:** 6
**Confidence:** 3

**Review:**

The authors provide a natural definition of adversarial examples for natural language transduction (meaning-preserving on source side while meaning-destroying on target side) and a human judgment task to measure it. They then investigate three different ways of generating adversarial examples and show that a metric based on character n-gram overlap (chrF) has a stronger correlation with human judgment. Finally, they show that adversarial training with the attack most consistent with the introduced meaning-preservation criteria results in improved robustness to this type of attack without degradation in the non-adversarial setting.

Overall this is a strong paper. It is well structured, the problem studied is highly interesting and the proposed meaning-preserving criteria and human judgement will be useful to anyone interested in adversarial attacks for natural language. While the studied attack methods are fairly primitive, the empirical results are still interesting.

Comments
---------------
I wish the authors would include experiments with CharSwap where OOV is not forced as I'm not sure the assumption that OOV is more meaning-destroying in the target side is necessarily true (one could also argue that since the models are already trained with OOV words, they may be more robust to OOV words than in-vocabulary words in the wrong context).

It would be nice to add correlation for each type of constraint as well to Table 2. The result would be even stronger if the experiment was replicated in the opposite direction or for another language pair as well.

I don't understand why the adversarial output in the second example in table 4 has a RDchrF of zero (the word July is completely dropped).

From Table 6 it looks like random sampling is actually slightly better than adversarial training in terms of robustness to CharSwap attacks in the Transformer model. Moreover, the benefit of adversarial rather than random sampling is quite small in the LSTM model as well. This could be made more clear in the text.

It would be interesting to see how adversarial training with the CharSwap method fares against the unconstrained and kNN attacks in table 6.

---

> ### Author Response · Authors · 2018-11-09
> **Author response for reviewer 2**
>
> We thank the reviewer for their encouraging comments. We appreciate the importance that they put on the problematic of evaluating meaning preservation in adversarial attacks.
>
> Regarding some specific comments:
>
> > What about CharSwap where OOV is not forced?
>
> This would indeed be an interesting experiment, which we will try to carry-out should time permit. Note that an attractive property of OOV is that it renders the optimization problem (2) relatively simple which might favor gradient-based attacks. Finally, as the reviewer pointed out, the effectiveness of the various attacks is not the focal point of the paper.
>
> > It would be nice to add correlation for each type of constraint as well to Table 2
>
> For each constraint:
>
> Unconstrained:
>  - BLEU: 0.582
>  - METEOR: 0.572
>  - chrF: 0.599
> kNN:
>  - BLEU: 0.533
>  - METEOR: 0.584
>  - chrF: 0.606
> CharSwap:
>  - BLEU: 0.273
>  - METEOR: 0.318
>  - chrF: 0.382
>
> As the reviewer can see, we observe the same trend for each kind of constraints (BLEU<meteor<chrF), except for Unconstrained where all metrics correlate highly with human judgment. However, none of those differences are statistically significant (with p<0.01). Note that the sample size is also smaller.
>
> We will include these results in a revised version of the paper.

---

> > ### Comment · AnonReviewer2 · 2018-11-12
> > **Modified rating in light of other reviews**
> >
> > I still like the overall mission of this paper and found it highly readable. However, after a more careful reading I do agree with the issues raised by the other reviewers. It seems that there is a fundamental question in the field as to a) how important meaning preservation is for adversarial attacks and b) how this should be assessed. In its current form, I don't think this paper provides satisfactory answers to these questions, but it does point at an important topic to be resolved.

---

> > > ### Author Response · Authors · 2018-11-24
> > > **Response to reviewer follow-up**
> > >
> > > We thank the reviewer for their follow-up. Regarding issues raised by other reviewers, we have provided additional follow-up comments and we invite the reviewer to consult them.

---

### Official Review · AnonReviewer1 · 2018-11-03
**Interesting, but significant methodological and experimental problems.**

**Rating:** 3
**Confidence:** 4

**Review:**

Summary: Proposes a framework for performing adversarial attacks on an NMT system in which perturbations to a source sentence aim to preserve its meaning, on the theory that an existing reference translation will remain valid if this is done. Given source and target metrics for measuring similarity, an attack is deemed successful if the source difference is smaller than the relative decrease in target similarity to the reference. A first experiment measures correlation with human judgements of similarity between original and perturbed sentences, and concludes that chrF is better than BLEU and METEOR for this purpose. Next, standard gradient-based adversarial attacks are carried out, replacing the three tokens that result in the biggest drop in (approximate) reference probability, either 1) with no constraints, 2) constrained to character swaps of the original token, or 3) constrained be among the 10 closest embeddings to the original token. In comparisons on three language pairs from IWSLT,  the constrained attacks are found to preserve meaning and yield more successful attacks according to the current framework. The Transformer architecture was also found to deal less well with attacks under the 10-closest embedding constraint. Finally, adversarial training with the character-swap constraint confers some robustness to this attack, without degrading performance on normal text.

I think it is a good idea to formalize a method for carrying out and assessing adversarial attacks, but the framework proposed here seems too narrow, as it excludes adversarial inputs that are sensible but not a close perturbation of an existing source/reference pair, or ones that contain varying amounts of noise. It is more difficult to measure output quality for such attacks, but that doesn’t seem like a good reason for excluding them from what is intended to be a general framework. Note also that “more difficult” doesn’t mean impossible, since good attacks can produce severely degraded output that is relatively easy to detect.

I found some of the methodology questionable. Limiting source perturbations to character swaps and neighbors in embedding space, then using automatic metrics to measure semantic distance seems both unnecessary and unlikely to succeed. Unnecessary because knowing the class of perturbation already gives you a lot of information about semantic distance. Unlikely to succeed because automatic metrics are too coarse to reliably distinguish among different perturbations. This is particularly obvious in the case of using character ngram distance (chrF) to determine which character swaps preserve meaning best. The experiments that support the viability of automatic metrics in 4.2 do so by measuring correlation with human judgment when the number of perturbed tokens varies from 1 to 3. I think the good correlation is likely due to the metrics being able to detect that, eg, changing 3 tokens makes things worse than changing only one. To be convincing, the experiments would have to be repeated with number of perturbations fixed at 3, to match the setting in the remaining experiments.

Apart from the interesting observation about the Transformer’s performance on embedding-neighbor attacks mentioned above, it is difficult to know what conclusions to draw from the experiments. In 4.3 it seems obvious a priori that perturbations intended to be relatively meaning preserving would indeed preserve meaning better than unconstrained ones. Similarly, it is not surprising that character swaps that by design produce an OOV token will cause more damage than choosing a near neighbor in embedding space. In 5.3, training with OOVs (resulting from character swaps) is of course not likely to hurt performance on test sets containing few OOVs, and, as is known from previous work, it will improve robustness to the same kind of noise. A final comment about the experiments is that word-based systems are not state of the art, and it isn’t clear how much we could expect any conclusions to carry over to sub-word models.

To conclude, although this is an interesting initiative, both the framework and the methodology need to be tightened up.

Details:

End of 2.1: this would be easier to interpret if you had previously specified the allowed range for s_src.

3.2 For kNN, being semantically related doesn’t imply that the relationship is synonymy, as would be required for meaning preservation. It also doesn’t imply that the substitution will be grammatical, which could jeopardize meaning preservation even if the words are synonyms.

CharSwap seems odd. If you’re just going to replace a work with an OOV symbol in any case, why go to the trouble of swapping characters? No matter what actual semantic shift is caused by the swap, the model will always see exactly the same representation.

4.1 “Following previous work on adversarial examples for seq2seq models (Belinkov & Bisk, 2018; Ebrahimi et al., 2018a)” - this is misleading: Ebrahimi et al only work with classification, and don’t use IWLST.

4.1 Should mention the size of the training sets in this section.

Table 1, first sentence, CharSwap example omits “faire”.

4.3, “Adding Constraints Helps Preserve…” last sentence: but here you need to reason in the opposite direction.

5.2 It would be good to also give absolute scores for table 6, so we can judge how much the systems actually benefited, and whether these gains were statistically significant.

---

> ### Author Response · Authors · 2018-11-09
> **Author response for reviewer 1 (Pt 1)**
>
> We thank the reviewer for their extensive and in-depth review, and glad that the overall direction was deemed interesting and valuable, even if there were disagreements with the experimental details. We believe that a number of these disagreements have already been resolved in the paper, or can be resolved with additional experimentation, which we will try our hardest to do. Please see the detailed responses below, and we are happy to address any additional comments.
>
> > The framework is too narrow, doesn’t consider adversarial inputs that are not perturbations of existing samples
>
> The reviewer is correct that our contribution focuses on adversarial perturbations only. We do not think that this setting is too narrow as those kinds of attacks constitute a significant chunk of the literature on the topic (in NLP and other areas [1,2,3,4] inter alia).
>
> > The framework excludes perturbations with varying amounts of noise
>
> On the contrary, our framework implicitly quantifies the amount of noise through the value of the semantic similarity metric. For example adversarial perturbations within edit distance 3 will have lower eg. BLEU score than perturbations within edit distance 1. Ultimately this depends on the chosen similarity metric.
>
> > Perturbations are limited to nearest neighbors and character swap
>
> Please keep in mind that for both the human judgment experiments we include unconstrained perturbations as well, as explained in 4.2.
>
> > kNN and charSwap constraints are unnecessary because knowing the class of perturbation already gives you a lot of information about semantic distance
>
> This is certainly somewhat true, but there are exceptions. For example, a nearest neighbor may be syntactically similar but semantically distant, or swapping characters may change the word to another meaning: “care” -> “acre”. However, this is irrespective of our main point here, which is to show that meaning-preservation *should* be evaluated independently of what a-priori knowledge one has of the level of meaning-preservation.
>
> > Automatic metrics are too coarse to reliably distinguish among different perturbations
>
> Results in Table 3 seem to contradict this statement. However, we do understand that our metrics are not perfect, and future metrics may make the results even more significant.
>
> > I think the good correlation is likely due to the metrics being able to detect that, eg, changing 3 tokens makes things worse than changing only one
>
> First, we would like to point out that this does not explain the (statistically significant) difference in correlations between eg. BLEU and RDchrF in the source. However the reviewer raises an interesting point. We computed correlations with each edit-distance bin and the results are as follow:
>
> Edit distance 1:
>  - BLEU = 0.351
>  - METEOR = 0.351
>  - chrF = 0.486
> Edit distance 2:
>  - BLEU = 0.403
>  - METEOR = 0.424
>  - chrF = 0.588
> Edit distance 3:
>  - BLEU: 0.334
>  - METEOR: 0.392
>  - chrF: 0.559
>
> In summary, our conclusions hold within each edit distance category (chrF is better than BLEU and meteor with p<0.01, with the caveat that the sample size is now smaller for each subset). Therefore the good correlation is not due only to the metrics being able to detect different edit distances.
>
> We will add the results to the revised version of the paper.
>
> > The conclusions are not clear
>
> We will try to clarify this in the paper. The null hypothesis here is that no one type of adversarial attack is better than the other at preserving meaning, and therefore meaning-preservation should not be evaluated. Our experiments show that this is not the case, and the choice of adversarial attack highly affects the amount that meaning is preserved. Thus, when a new variety of adversarial attack is conceived, meaning preservation should definitely be taken into account when comparing it to previous attacks.
>
> > it seems obvious a priori that perturbations intended to be relatively meaning preserving would indeed preserve meaning better than unconstrained ones
>
> We agree with the reviewer that this appears obvious, particularly in hindsight, but the previous literature has not taken this into account in their evaluations whatsoever. The attacks compared in this paper are relatively straightforward and our conclusions are logical, but we expect that for future works that propose more sophisticated attacks we may not be able to predict the conclusions a-priori nearly as easily. Our point is that future work in the literature should take this problem into account when performing evaluation.

---

> > ### Author Response · Authors · 2018-11-09
> > **Author response for reviewer 1 (Pt 2)**
> >
> > > training with OOVs (resulting from character swaps) is of course not likely to hurt performance on test sets containing few OOVs
> >
> > This is a reasonable remark, although a discussion could be had as to whether changing the training distribution while keeping the test distribution and model capacity constant should not decrease test performance in general.
> >
> > > word-based systems are not state of the art, and it isn’t clear how much we could expect any conclusions to carry over to sub-word models
> >
> > We acknowledge that this is a valid criticism of our work. Although we expect that our central contribution (clarifying the importance of evaluating meaning-preservation in adversarial perturbations) will carry over to sub-word models, we will be running experiments during the rest of the rebuttal period to validate this claim.
> >
> > > For kNN, being semantically related doesn’t imply that the relationship is synonymy
> >
> > We agree with the reviewer, however
> > (1): this is a somewhat good approximation in languages where we may not have access to precise synonymy information (like wordnet)
> > (2): Our point is precisely that even though one may have preconceptions of the capacity of a class of perturbations to preserve meaning, meaning-preservation should still be evaluated explicitly.
> >
> > > If you’re just going to replace a work with an OOV symbol in any case, why go to the trouble of swapping characters?
> >
> > For for human and automatic evaluation, we still need to provide “valid” sentences that don’t just replace words with “<unk>”. This is a quirk of word-based model and our experiments with sub-word models should help resolve this.
> >
> > > Ebrahimi et al only work with classification, and don’t use IWLST
> >
> > We suspect the reviewer is referring to the Ebrahimi et al 2018b Hotflip paper, however the 2018a reference points to the COLING paper “On adversarial examples for character-level neural machine translation” paper which indeed works with MT on IWSLT. Arguably the lettering is a bit confusing here, we will address this in a revised version.
> >
> > References:
> > [1]: Goodfellow, Ian J., Jonathon Shlens, and Christian Szegedy. "Explaining and Harnessing Adversarial Examples." arXiv preprint arXiv:1412.6572 (2014).
> > [2]: Ebrahimi, Javid, et al. "Hotflip: White-box adversarial examples for text classification." Proceedings of the 56th Annual Meeting of the Association for Computational Linguistics (Volume 2: Short Papers). Vol. 2. 2018.
> > [3]: Cheng, Minhao, et al. "Seq2Sick: Evaluating the Robustness of Sequence-to-Sequence Models with Adversarial Examples." arXiv preprint arXiv:1803.01128 (2018).
> > [4]: Ebrahimi, Javid, Daniel Lowd, and Dejing Dou. "On Adversarial Examples for Character-Level Neural Machine Translation." Proceedings of the 27th International Conference on Computational Linguistics. 2018.

---

> > > ### Comment · AnonReviewer1 · 2018-11-12
> > > **Still not convinced by the framework and the experiments**
> > >
> > > I appreciate the detailed and careful responses by the authors, but I feel that they don’t directly address the main concerns I had with this paper. I have tried to restate these more clearly below.
> > >
> > > Regarding the proposed framework, I don’t think it’s a good idea to try to limit the scope of adversarial attacks to ones that are “meaning preserving”, for several reasons. First the notion is hard to define, especially when perturbations produce ill-formed input. For instance, does introducing a nonsense token at the beginning of a sentence preserve its meaning? This is not just a theoretical question, since such perturbations occur in real data, and can trigger “hallucinatory” behavior in NMT that is very different from what a human translator would do. Second, even if we had a satisfactory definition for “meaning preserving” in this context, it would be very difficult to measure reliably. This is essentially the problem of paraphrase, and it’s not any easier than MT - in fact, harder in practice, due to the lack of parallel data. Finally, even if the above two problems were resolved, I don’t see the point in specifically excluding attacks that change meaning. On the contrary, changing words or grammatical attributes in constrained ways seems like very fertile ground to explore. For instance, “John loves Mary” -> “Bob loves Mary”, “John sees Mary”, “John loved Mary”, “John is loved by Mary”, etc. Of course, these would invalidate any existing reference translation, but permissible changes to the reference could be checked automatically if the experiment were set up carefully. There is work on this from the burgeoning field of challenge sets for MT; see, eg, the “extra test suites” from WMT 2018. Furthermore, cases where the attack triggers hallucinatory behavior are relatively easy to detect, even without a reference. Such behavior is perhaps the most significant problem for MT robustness at the moment, and it is absent from the current paper.
> > >
> > > Turning to what the paper actually does, the basic idea to measure the discrepancy between source-side and target-side semantic difference associated with an adversarial attack makes sense in principle. In practice, given the current state of the art, such measurements are always going to amount to just surface distances like chrF as espoused here. If what we’re really doing is perturbing some characters in the source and measuring how many characters change in the target as a result, it seems clearer just to describe it that way. Absent careful constraints like limiting perturbations to word-internal swaps, many such changes won’t preserve meaning, but as I argue above, that’s not necessarily a bad thing.
> > >
> > > A final note about the central character-swap experiments. The technique is to find the three tokens that result in the biggest probability drop when replaced with OOVs (resulting from character swapping), then measure the resulting target-side relative delta chrF. That’s fine, although it’s not clear what is to be gleaned from the results. What’s not fine is to also measure the source-side chrF and compare this to the target-side chrF.  From the model’s perspective, all these perturbations are exactly the same (three OOVs, regardless of how they were produced), so the relation between source- and target-side chrF is completely arbitrary. Even from a human perspective, the source chrF scores are unlikely to be meaningful. As the authors correctly observe, the vast majority of word-internal character swaps are meaning preserving in the sense that we automatically correct them when we read them in context. So the role of chrF here is to distinguish between fine degrees of preserving meaning, a task that seems well out of reach for raw character ngrams.

---

> > > > ### Author Response · Authors · 2018-11-24
> > > > **Response to reviewer follow-up**
> > > >
> > > > We extend our thanks to the reviewer for their detailed follow-up and the thoughtful discussion it is generating.
> > > > We will again attempt to summarize what we think are the reviewer’s main point and address these.
> > > >
> > > > > Meaning preservation is hard to define (example: what about adding a nonsense token?)
> > > >
> > > > In the specific example given, we would refer to human judgement. Note that the case of ill-formed inputs is taken into account in the rating scale we propose in section 2.2.1 (specifically options 0 and the distinction between options 4 and 5).
> > > >
> > > > > Evaluating meaning preservation is very hard (essentially paraphrase detection), at least as hard as MT.
> > > >
> > > > While this is certainly true, our general framework does not rely on the existence of a perfect model of semantic equivalence to be of use. Indeed, we show in the paper that, while not ideal, an automatic metric still provides a positive signal to differentiate between eg. unconstrained attacks (which are not expected to be meaning-preserving) and charswap attacks (which are expected to preserve meaning to some extent). This is useful because in cases where meaning preservation is suspected, but less straightforward (our kNN constraint for example), we can only rely on evaluation a posteriori, and chrF provides a good proxy.
> > > >
> > > > > Why focus on meaning-preserving attacks? What about cases where we alter the meaning of the source, but also alter the reference accordingly?
> > > >
> > > > While we think that the setting(s) described by the reviewer are highly relevant to adversarial attacks and MT in general, they fall out of the intended scope of this paper, which is adversarial perturbation where the resulting output is compared to the reference. So while our framework doesn’t cover the entirety of the area of adversarial attacks on MT (let alone MT robustness), we think it is still relevant for a non-negligible part of the literature (cf references in the paper, notably the last paragraph of Section 6).
> > > >
> > > > > If what we’re really doing is perturbing some characters in the source and measuring how many characters change in the target as a result, it seems clearer just to describe it that way
> > > >
> > > > We don’t think that this description of the setting is completely accurate, as it leaves out perturbations where we change entire words (Unconstrained and kNN) in the source.
> > > >
> > > > > Since the model sees CharSwaped words as OOVs no matter how they were perturbed, the relationship between source chrF and target RDchrF is arbitrary (as source chrF can vary depending on the perturbation method while target RDchrF doesn’t change).
> > > >
> > > > We argue that in MT, preprocessing (including replacing OOVs with a special token) is part of the model. From this perspective, an attack that would introduce 3 OOVs, but obtain these OOVs by eg. replacing words with nonsensical sequences of characters will not be the same as our CharSwap attack from the model’s point-of-view.
> > > >
> > > > > Source chrF are unlikely to be meaningful (to humans). The role of chrF here is to distinguish between fine degrees of preserving meaning, a task that seems well out of reach for raw character ngrams.
> > > >
> > > > In the context of CharSwap, the role of chrF (or any other metric) is not to distinguish between character swaps that are meaning preserving or not. Rather, it is to distinguish between types of constraints (eg. CharSwap vs Unconstrained).

---

> > > > > ### Comment · AnonReviewer1 · 2018-11-26
> > > > > **Still disagree**
> > > > >
> > > > > I continue to stand by my original review. I think this effort is ambitious and interesting, but the limitation to preserving meaning makes the problem both unnecessarily difficult and insufficiently broad to be of practical utility at present. I don’t think the experiments carried out within this framework are very informative.
> > > > >
> > > > > On the specific question of source-side perturbations that always produce OOVs, I can’t see that redefining ‘model’ makes any difference. The point is just that if the translation process is unaffected by a particular class of source perturbations, as in your experiments, it makes no sense to compare these to the resulting target-side perturbations.

---

### Official Review · AnonReviewer3 · 2018-11-04
**Not enough novelty for acceptance**

**Rating:** 4
**Confidence:** 4

**Review:**

The paper is about meaning-preserving adversarial perturbations in the context of Seq2Seq models. The paper proposes two ways of achieving that: (a) kNN - substituting word with nearest neighbors from the word embedding space, and (b) character swapping. It's debatable if character swapping is really meaning preserving since a lot of typos can really change the word. Similarly a case can be made about kNNs as well. But even if these are the best approximations we have, I have some major issues about the novelty of the work. Firstly, while the authors are trying to pitch the work in a new mold, there's major overlap with Belinkov and Bisk, 2018. The use of character swapping as an adversarial perturbation/noise and the subsequent benefits of training with adversarial noise have already been shown in Belinkov and Bisk, 2018. Secondly, the models tested are operating at word-level whereas most of the state-of-the-art systems nowadays are all using subword-level vocabularies. The character swap method presented would need to be adapted and some of the takeaways from results are hence less relevant for the current SOTA models. Coming to positives, the two real contributions for me are: (a) the result that chrF correlates better with human judgement, and (b) the measurement of adversarial perturbation's success measured via a sum that includes relative decrease in target score and the similarity of source sentence with the perturbed version. However, these are minor contributions and not enough to cover up the major flaws that I discussed above.

Some other minor issues:
(a) Table 1: The first example has the CharSwap row missing the word "faire".
(b) Section 3.1.1: "d" is not defined when discussing time complexity.
(c) No separate section 3.1.2 required as it can be merged with 3.1.1 and would be more easy to understand without confusing the readers that there's some context change.
(d) Table 6 entries are not clearly defined. How is robustness measured?

---

> ### Author Response · Authors · 2018-11-09
> **Author response for Reviewer 3**
>
> We thank the reviewer for their time and their comments.
>
> Before addressing specific comments, we would like to emphasize that the intended contribution of this paper is not so much about proposing new adversarial attacks as to raise the issue of explicitly evaluating meaning preservation in the context of adversarial attacks on sequence to sequence models. We respectfully disagree with the reviewer that this is a minor contribution, as there is a flourishing literature on adversarial attacks on NLP (and seq2seq) models that often sidesteps this important issue [1,2,3].
>
> Now on to specific remarks:
>
> > It is debatable whether kNN or CharSwap are indeed preserving meaning
>
> We agree and the point of this work is precisely to show that this meaning preservation should not just merely be left as an assumption but actually evaluated via human judgement or automatic proxies thereof.
>
> > Major overlap with Belinkov & Bisk (2017)
>
> We disagree with this assessment, While there are similarities in the choice of perturbations (notably CharSwap), B&B only look at random character replacements whereas we use a systematic approach to generate perturbations using gradients. Moreover, their contribution focuses on the brittleness of character level MT systems to noise, while ours is about the necessity to evaluate the level of meaning preservation of any kind of perturbation. As such, we think that these two works present very distinct contributions.
>
> > Word level models whereas SOTA models use subwords (BPE)
>
> This is a fair criticism that has been raised by several reviewers. To clarify, we expect our main contribution (evaluating meaning-preservation is important) to carry over to subwords (or character models). We will be running experiments on BPE models to confirm this hypothesis before the end of the author response period. We would like to emphasize here again that the specific constraints are not the main contribution of the paper.
>
> > minor issues
>
> We acknowledge these and will address those in the revised version. Specifically for (b):  d is the dimension of word embeddings and for (d):  the metric is RDchrF
>
> References:
> [1]: Zhao, Zhengli, Dheeru Dua, and Sameer Singh. "Generating natural adversarial examples." arXiv preprint arXiv:1710.11342 (2017).
> [2]: Cheng, Minhao, et al. "Seq2Sick: Evaluating the Robustness of Sequence-to-Sequence Models with Adversarial Examples." arXiv preprint arXiv:1803.01128 (2018).
> [3]: Ebrahimi, Javid, Daniel Lowd, and Dejing Dou. "On Adversarial Examples for Character-Level Neural Machine Translation." Proceedings of the 27th International Conference on Computational Linguistics. 2018.

---

> > ### Comment · AnonReviewer3 · 2018-11-19
> > **Still not convinced**
> >
> > Thank you for such a detailed feedback.
> >
> > I agree with the authors that in the light of their focus on meaning preserving adversarial perturbations for NMT, their work is indeed novel. However, there are certain issues with the approach proposed which have been raised by other reviewers as well:
> > (a) Measuring semantic similarity is an extremely hard problem in itself.
> > (b) The correlation of chrF with human judgement doesn't inspire much confidence, especially given that it might be already inflated because of the varying number of perturbations introduced.
> >
> > Due to these fundamental issues, the framework proposed is not convincing. Lack of subword-based model experimentation also make the experimental section weak. Hence I'll keep my original rating.

---

> > > ### Author Response · Authors · 2018-11-24
> > > **Response to reviewer follow-up**
> > >
> > > We thank the reviewer for following up. We have addressed some of the points raised by other reviewers in separate replies, let us summarize and reiterate these responses here for the sake of clarity:
> > >
> > > > (a) Measuring semantic similarity is an extremely hard problem in itself
> > >
> > > While this is certainly true, our general framework does not rely on the existence of a perfect model of semantic similarity to be of use. Indeed, we show in the paper that, while not ideal, an automatic metric still provides a positive signal to differentiate between e.g. unconstrained attacks (which are not expected to be meaning-preserving) and charswap attacks (which are expected to preserve meaning to some extent). This is useful because in cases where meaning preservation is suspected, but less straightforward (our kNN constraint for example), we can only rely on evaluation a posteriori, and chrF provides a good proxy.
> > >
> > > > (b) The correlation of chrF with human judgement doesn't inspire much confidence, especially given that it might be already inflated because of the varying number of perturbations introduced.
> > >
> > > First, we would like to reiterate that the main takeaway from the human judgement experiments is not so much the absolute value of the correlation coefficient---although of course it is important that there be a positive correlation. Rather the difference between different metrics (BLEU, METEOR, chrF) is of greater interest.
> > > That being said, we looked up correlations within each edit-distance bin and the results are as follow:
> > >
> > > Edit distance 1:
> > > BLEU = 0.351
> > > METEOR = 0.351
> > > chrF = 0.486
> > > Edit distance 2:
> > > BLEU = 0.403
> > > METEOR = 0.424
> > > chrF = 0.588
> > > Edit distance 3:
> > > BLEU: 0.334
> > > meteor: 0.392
> > > chrF: 0.559
> > >
> > > In summary, our conclusions hold within each edit distance category (chrF is better than BLEU and meteor with p<0.01, with the caveat that the sample size is now smaller for each subset). Therefore the good correlation is not due only to the metrics being able to detect different edit distances (=number of perturbations).
> > >
> > > If, this point being addressed, the reviewer still thinks that the correlation of chrF with human judgement does not inspire confidence, we would be grateful if they could elaborate on their concerns further, so that we can either dispel them or ameliorate the experimental setup.

---

### Official Review · AnonReviewer4 · 2018-12-10
**Interesting framework but lack of novelty and unclear evaluation of attack**

**Rating:** 4
**Confidence:** 4

**Review:**

The authors present a framework for creating meaning-preserving adversarial examples, and give two methods for such attacks. One is based on k-nn in the word embedding space, and another is based on character swapping. The authors further study a series of automatic metrics for determining whether semantic meaning in the input space has changed, and find that the chrF method produces scores most correlated with human judgement of semantic meaning. The authors finally give an evaluation of the two methods.


Positive:
- The authors give a framework with the explicit goal of preserving meaning in attacks.

Negative:
- Unclear novelty: previous work also gives the goal of preserving input meaning in attacks, even if the attacks themselves do not preserve meaning effectively (ie Zhao et al)
- Unclear attack effectiveness: The chrF scores for CharSwap and kNN methods have higher chrF scores than the "unconstrained" method, but it is unclear what this means in context. Similarly, the RDchrF scores show that the average output changes in meaning by some amount, but the authors do not show in context what this really means in terms of meaning.

Details of negatives:
Unclear attack effectiveness:
- Using chrF score as a proxy for human judgement is unmotivated. There is little analysis of the distribution of chrF scores compared to human judgement - the only analysis given is that a) there is a .586 correlation on French and .497 correlation on English, and b) that :"90% of French sentence pairs to which humans gave a score of 4 or 5 in semantic similarity have a chrF > 78". It would be good to plot the distribution of chrF score vs human judgement, so that the reader is able to tell what the chrF scores really mean in context here - a correlation score of approximately .5 is difficult to interpret.
- The chrF/RDchrF scores in the source and target spaces (respectively) as they relate to "meaning-preservingness" suffer from uninterpretability as a reader, both because of the point above and also because there are few examples of adversarial examples with their chrF/RDchrF scores given (only two).

---

### Meta-Review · Area_Chair1 · 2018-12-16
**Important goal but the evaluation and relationship to the previous work needs improvement**

**Confidence:** 4
**Recommendation:** Reject

**Metareview:**

This paper present a framework for creating meaning-preserving adversarial examples. It then proposes two attacks within this framework: one based on k-NN in the word embedding space, and another one based on character swapping.

Overall, the goal of constructing such meaning-preserving attacks is very interesting. However, it is unclear how successful the proposed approach really is in the context of this goal.

Additionally, it is not clear how much novelty there is compared to already existing methods that have a very similar aim.